# SENP3 maintains the stability and function of regulatory T cells via BACH2 deSUMOylation

Xiaoyan Yu[1], Yimin Lao[2], Xiao-Lu Teng[1], Song Li[1], Yan Zhou[1], Feixiang Wang[1], Xinwei Guo[1], Siyu Deng[1], Yuzhou Chang[1], Xuefeng Wu[1], Zhiduo Liu[1], Lei Chen[1], Li-Ming Lu[1], Jinke Cheng[2], Bin Li[1], Bing Su[1], Jin Jiang[3], Hua-Bing Li [1], Chuanxin Huang[1], Jing Yi[2] & Qiang Zou[1]

Regulatory T (Treg) cells are essential for maintaining immune homeostasis and tolerance, but the mechanisms regulating the stability and function of Treg cells have not been fully elucidated. Here we show SUMO-specific protease 3 (SENP3) is a pivotal regulator of Treg cells that functions by controlling the SUMOylation and nuclear localization of BACH2. Treg cell-specific deletion of *Senp3* results in T cell activation, autoimmune symptoms and enhanced antitumor T cell responses. SENP3-mediated BACH2 deSUMOylation prevents the nuclear export of BACH2, thereby repressing the genes associated with CD4$^+$ T effector cell differentiation and stabilizing Treg cell-specific gene signatures. Notably, SENP3 accumulation triggered by reactive oxygen species (ROS) is involved in Treg cell-mediated tumor immunosuppression. Our results not only establish the role of SENP3 in the maintenance of Treg cell stability and function via BACH2 deSUMOylation but also clarify the function of SENP3 in the regulation of ROS-induced immune tolerance.

---

[1] Shanghai Institute of Immunology, Department of Immunology and Microbiology, Key Laboratory of Cell Differentiation and Apoptosis of Chinese Ministry of Education, Shanghai Jiao Tong University School of Medicine, 280 South Chongqing Road, Shanghai 200025, China. [2] Shanghai Key Laboratory for Tumor Microenvironment and Inflammation, Key Laboratory of Cell Differentiation and Apoptosis of Chinese Ministry of Education, Department of Biochemistry and Molecular Cell Biology, Shanghai Jiao Tong University School of Medicine, 280 South Chongqing Road, Shanghai 200025, China. [3] Department of Molecular Biology, University of Texas Southwestern Medical Center, Dallas, TX 75390, USA. These authors contributed equally: Xiaoyan Yu, Yimin Lao, Xiao-Lu Teng. Correspondence and requests for materials should be addressed to H.-B.L. (email: huabing.li@shsmu.edu.cn) or to C.H. (email: huangcx@shsmu.edu.cn) or to J.Y. (email: yijing@shsmu.edu.cn) or to Q.Z. (email: Qzou1984@sjtu.edu.cn)

Regulatory T (Treg) cells play a central role in the maintenance of peripheral immune tolerance and homeostasis[1,2]. These cells can also strongly dampen antitumor T cell immune responses, thereby decreasing the efficacy of tumor immune surveillance[3]. The key transcription factor Foxp3 has a critical role in the differentiation and function of Treg cells[4,5]. Impaired Foxp3 expression attenuates the immunosuppressive capacity of Treg cells, which is linked to severe autoimmune diseases[6]. In addition to the master transcription factor Foxp3, various transcription factors repress effector T (Teff) cell transcriptional programs and maintain Treg cell-specific gene signatures. For example, Musculin (MSC) is critical for the induction of Treg cells via the suppression of the T helper (Th)-2 cell-specific transcriptional program[7]. Likewise, BACH2 is required for repressing effector programs in the maintenance of Treg cell-mediated immune homeostasis[8,9]. Therefore, the function and stability of Treg cells are tightly controlled by transcriptional programs.

SUMOylation is an important reversible post-translational protein modification[10]. DeSUMOylation is catalyzed by SUMO-specific proteases (SENPs)[11]. SUMOylation plays a functional role in the regulation of activities of specific transcription factors by mediating protein stability, nuclear transport, recruitment of chromatin remodeling machinery or transcriptional regulation[12–14]. It has been reported that SUMOylation is essential for T cell activation and differentiation. For example, T cell antigen receptor (TCR)-induced SUMO1 conjugation of PKC-θ is required for effective T cell activation[15]. T cell-specific SUMO2-overexpressing transgenic mice exhibit enhanced generation and function of interleukin (IL)-17-producing CD8$^+$ T cells[16]. The loss of SUMO-conjugating enzyme UBC9 inhibits Treg cell expansion and function, leading to severe autoimmune diseases[17]. However, it is still unknown whether SENP-mediated deSUMOylation regulates transcriptional programs in different types of immune cells, especially in Treg cells.

The SUMO2/3-specific protease SENP3 is sensitive to the increase in reactive oxygen species (ROS). ROS can stabilize SENP3 by blocking its ubiquitin-mediated degradation[18,19]. Although the physiological role of SENP3 in immune responses is largely unclear, ROS have been demonstrated to have a protective role in immune-mediated diseases. A lack of ROS has been associated with increased susceptibility to autoimmunity and arthritis, coupled with enhanced T cell responses[20]. In contrast, increased ROS levels have been shown to attenuate experimentally induced asthmatic inflammation and colitis[21]. Additionally, elevated ROS can suppress immune responses in the tumor microenvironment, which contributes to tumor-induced immunosuppression[22,23]. Indeed, reduced ROS levels impair Treg cell function[24], but the underlying molecular mechanism is still unknown. Thus, it is of interest to determine whether SENP3 is a critical regulator of ROS-induced immune tolerance.

In this study, we show that SENP3 specifically regulates Treg cell stability and function by promoting BACH2 deSUMOylation, which in turn prevents the nuclear export of BACH2 to repress Teff cell-transcriptional programs and maintain Treg cell-specific gene signatures. SENP3 rapidly accumulates in Treg cells following TCR and CD28 stimulation in a ROS-dependent manner. Further pharmacological approaches indicate that the loss of ROS attenuates Treg cell-mediated immunosuppression and enhances antitumor T cell responses. These findings identify SENP3 as an important regulator of Treg cell-specific transcriptional programs via BACH2 deSUMOylation and suggest that SENP3 mediates the regulation of Treg cell function by ROS.

## Results

**SENP3 functions in T cells to maintain immune homeostasis.** To assess the function of SENP3 in immune cells, we first analyzed its expression at the protein level and found that SENP3 was highly expressed in T cells (Supplementary Fig. 1a). This prompted us to investigate the role of SENP3 in T cell function. To this end, we crossed *Senp3*-flox mice with *Cd4*-Cre mice to obtain *Senp3* T cell-conditional knockout (*Senp3*$^{fl/fl}$*Cd4*-Cre) mice (Supplementary Fig. 1b, c). The 6-week-old *Senp3*$^{fl/fl}$*Cd4*-Cre mice did not exhibit obvious abnormalities in thymocyte development or peripheral T cell frequency (Supplementary Fig. 1d, e). However, the proportion of activated or memory-like CD4$^+$ and CD8$^+$ T cells was substantially greater in the spleens of 8-week-old *Senp3*$^{fl/fl}$*Cd4*-Cre mice than in those of *Senp3*-wild-type (*Senp3*$^{+/+}$*Cd4*-Cre) mice (Fig. 1a). The 8-week-old *Senp3*$^{fl/fl}$*Cd4*-Cre mice had profoundly elevated IFN-γ-producing CD4$^+$ and CD8$^+$ Teff cell numbers in the spleen (Fig. 1b). This phenotype became more profound in 8-month-old *Senp3*$^{fl/fl}$*Cd4*-Cre mice (Fig. 1c, d). Consistent with the perturbed T cell homeostasis, infiltration of lymphocytes into the liver and lung was observed in 8-month-old *Senp3*$^{fl/fl}$*Cd4*-Cre mice (Fig. 1e). Although overall liver function, determined by the serum concentrations of alanine aminotransferase (ALT) and aspartate aminotransferase (AST), was normal in the 8-week-old *Senp3*$^{fl/fl}$*Cd4*-Cre mice (Supplementary Fig. 2a), it was abnormal in 8-month-old *Senp3*$^{fl/fl}$*Cd4*-Cre mice (Supplementary Fig. 2b). Moreover, the number of Teff cells in the liver and lung of 8-month-old *Senp3*$^{fl/fl}$*Cd4*-Cre mice was strikingly increased (Fig. 1f, g). Accordingly, *Senp3*$^{fl/fl}$*Cd4*-Cre mice displayed diminished survival compared to wild-type littermates (Fig. 1h). Collectively, these data demonstrated an important role for SENP3 in maintaining peripheral T cell homeostasis and preventing autoimmune responses.

**SENP3 ablation in Treg cells perturbs immune tolerance.** Since the hyperactivation of T cells can disrupt peripheral T cell homeostasis[25,26], we examined whether SENP3 deficiency enhances T cell activation. However, the induction of the early activation marker CD69 and the Teff cell marker CD44 was comparable between *Senp3*$^{+/+}$*Cd4*-Cre and *Senp3*$^{fl/fl}$*Cd4*-Cre CD4$^+$ T cells upon TCR and CD28 stimulation (Supplementary Fig. 3a). In addition, the loss of SENP3 did not alter the TCR-stimulated and CD28-stimulated production of cytokines, such as IL-2 and IFN-γ, in CD4$^+$ T cells (Supplementary Fig. 3b). Therefore, SENP3 deficiency does not influence T cell activation.

Treg cells function as key immunosuppressive cells to maintain T cell homeostasis[27–29]. Interestingly, a comparison of different CD4$^+$ T cell subsets generated in vitro revealed higher SENP3 expression in Treg cells (Supplementary Fig. 3c), and Foxp3 expression was not efficiently induced in the *Senp3*$^{fl/fl}$*Cd4*-Cre CD4$^+$ T cells compared with that in the *Senp3*$^{+/+}$*Cd4*-Cre CD4$^+$ T cells (Supplementary Fig. 3d). Therefore, we asked whether SENP3 deficiency in T cells affects Treg cell function. Indeed, the percentage and number of Treg cells in the thymus, spleen, and peripheral lymph nodes from 6-week-old *Senp3*$^{fl/fl}$*Cd4*-Cre mice were significantly lower than those from *Senp3*$^{+/+}$*Cd4*-Cre mice (Fig. 2a, b). We then generated chimeric mice by reconstituting *Rag1*$^{-/-}$ mice with a mixture of bone marrow (BM) cells from *Senp3*$^{fl/fl}$*Cd4*-Cre (CD45.2$^+$) mice and SJL (CD45.1$^+$) mice or *Senp3*$^{+/+}$*Cd4*-Cre (CD45.2$^+$) mice and SJL (CD45.1$^+$) mice. The frequency of *Senp3*$^{fl/fl}$*Cd4*-Cre Treg cells in the spleen was significantly lower than that of wild-type Treg cells (Supplementary Fig. 4a, b). Moreover, the perturbed T cell homeostasis appeared in chimeric mice reconstituted with a mixture of BM cells from *Senp3*$^{fl/fl}$*Cd4*-Cre mice and SJL mice (Supplementary

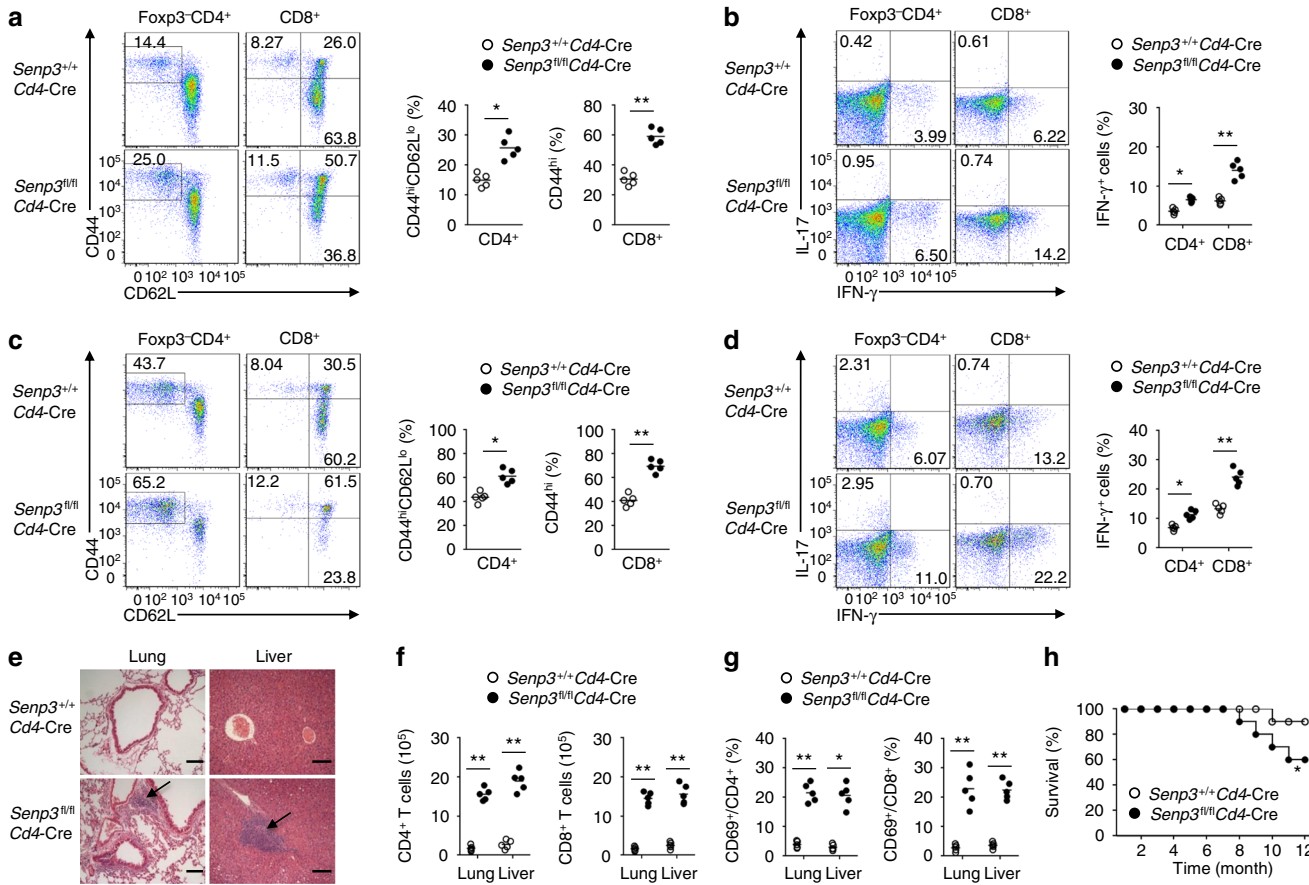

**Fig. 1** T cell-specific deletion of *Senp3* perturbs T cell homeostasis. **a** Flow cytometric analysis of the frequency of naive (CD44$^{lo}$CD62L$^{hi}$) and memory-like (CD44$^{hi}$CD62L$^{lo}$ for CD4$^{+}$ and CD44$^{hi}$ for CD8$^{+}$ T cells) CD4$^{+}$ and CD8$^{+}$ T cells in total splenocytes from 8-week-old *Senp3$^{+/+}$Cd4*-Cre and *Senp3$^{fl/fl}$Cd4*-Cre mice. **b** Flow cytometric analysis of the percentage of IFN-γ-producing and IL-17-producing CD4$^{+}$ and CD8$^{+}$ T cells in the spleen of 8-week-old *Senp3$^{+/+}$Cd4*-Cre and *Senp3$^{fl/fl}$Cd4*-Cre mice. **c, d** Flow cytometric analysis of the frequency of naive and memory-like (**c**) or IFN-γ-producing and IL-17-producing (**d**) T cells in total splenocytes from 8-month-old *Senp3$^{+/+}$Cd4*-Cre and *Senp3$^{fl/fl}$Cd4*-Cre mice. **e** Hematoxylin-eosin staining of the indicated tissue sections from 8-month-old *Senp3$^{+/+}$Cd4*-Cre and *Senp3$^{fl/fl}$Cd4*-Cre mice, showing immune cell infiltrations into the *Senp3$^{fl/fl}$Cd4*-Cre tissues (arrows). Bars, 100 μm. **f, g** Quantification of CD4$^{+}$ and CD8$^{+}$ T cells (**f**) and percentage of CD4$^{+}$ and CD8$^{+}$ T cells expressing CD69 (**g**) in the lungs and livers of 8-month-old *Senp3$^{+/+}$Cd4*-Cre and *Senp3$^{fl/fl}$Cd4*-Cre mice. **h** Survival curves of *Senp3$^{+/+}$Cd4*-Cre and *Senp3$^{fl/fl}$Cd4*-Cre mice (*n* = 10). Data are representative of three or more independent experiments. Error bars are the mean ± SEM values. *n* = 5 or 10. Two-tailed unpaired Student's *t* tests were performed. *$P < 0.05$; **$P < 0.01$

Fig. 4a, b). These data suggested that SENP3 ablation in Treg cells contributes to dysregulated immune tolerance.

To ascertain the Treg cell-intrinsic physiological relevance of SENP3 activity, we crossed *Senp3*-flox mice with *Foxp3*-Cre mice to generate *Senp3* Treg cell-conditional knockout mice (*Senp3$^{fl/fl}$Foxp3*-Cre), in which SENP3 was deleted in Treg cells (Supplementary Fig. 5a, b). Although the 6-week-old *Senp3$^{+/+}$Foxp3*-Cre and *Senp3$^{fl/fl}$Foxp3*-Cre mice had a similar percentage of thymocytes and peripheral T cells (Supplementary Fig. 5c, d), the *Senp3$^{fl/fl}$Foxp3*-Cre mice displayed a substantially lower frequency and number of Treg cells in the thymus, spleen and peripheral lymph nodes than did the *Senp3$^{+/+}$Foxp3*-Cre mice (Fig. 2c, d). Furthermore, the 8-week-old *Senp3$^{fl/fl}$Foxp3*-Cre mice exhibited an increased frequency of IFN-γ-producing T cells or IL-17-producing CD4$^{+}$ T cells in the spleen (Fig. 2e, f). Consistent with this, 8-month-old *Senp3$^{fl/fl}$Foxp3*-Cre mice had a concomitantly increased frequency of IFN-γ-producing T cells in the spleen (Supplementary Fig. 5e). In addition, 8-month-old *Senp3$^{fl/fl}$Foxp3*-Cre mice exhibited excessive infiltration of lymphocytes into the lung and liver (Fig. 2g). These observations further demonstrated that SENP3 ablation in Treg cells results in dysregulated immune tolerance.

**SENP3 is required for the suppressive function of Treg cells**. Treg cell-specific deletion of *Senp3* resulted in T cell activation and spontaneous autoimmune symptoms, which suggested that SENP3 is required for the suppressive function of Treg cells. To further confirm that conclusion, we examined the effect of Treg cell-specific SENP3 deletion on Treg cell function in vitro and in vivo. In vitro proliferation assays of responder T cells cocultured with Treg cells showed that SENP3 deletion impaired the suppressive activity of Treg cells (Fig. 3a). Using a well-characterized adoptive-transfer approach to measure the in vivo function of Treg cells[29], we found that the transfer of SENP3-deficient Treg cells together with the naive CD45RB$^{hi}$ CD4$^{+}$ T cells resulted in gradual weight loss (Fig. 3b), a greater frequency of memory and effector-like T cells (Fig. 3c) and hyperplasia of the colonic mucosa (Fig. 3d), but the transfer of wild-type Treg cells together with the naive CD45RB$^{hi}$ CD4$^{+}$ T cells did not (Fig. 3b–d). Therefore, SENP3 plays a critical role in Treg cell suppressive function.

The findings that SENP3 deficiency suppresses Treg cell function and promotes T cell activation indicated that targeting SENP3 might improve antitumor immunity. We next examined the role of Treg cell-specific deletion of SENP3 in regulating

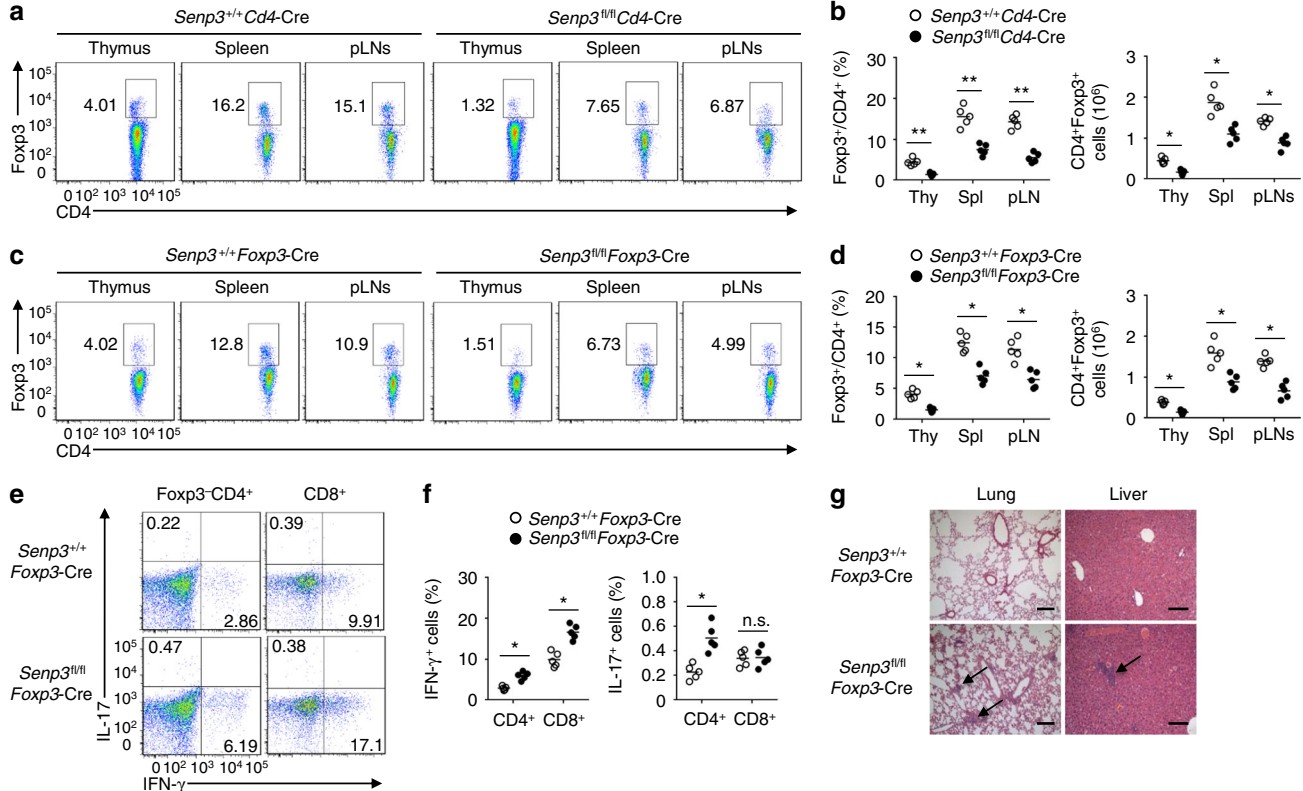

**Fig. 2** SENP3 ablation in Treg cells results in dysregulated immune tolerance. **a**, **b** Flow cytometric analysis of Treg cell percentage and number from the thymus (Thy), spleen (Spl), and peripheral lymph nodes (pLN) of 6-week-old $Senp3^{+/+}Cd4$-Cre and $Senp3^{fl/fl}Cd4$-Cre mice. **c**, **d** Flow cytometric analysis of Treg cell percentage and number from the thymus (Thy), spleen (Spl), and peripheral lymph nodes (pLN) of 6-week-old $Senp3^{+/+}Foxp3$-Cre and $Senp3^{fl/fl}Foxp3$-Cre mice. **e**, **f** Flow cytometric analysis of the percentage of IFN-γ-producing and IL-17-producing CD4+ and CD8+ T cells in the spleen of 8-week-old $Senp3^{+/+}Foxp3$-Cre and $Senp3^{fl/fl}Foxp3$-Cre mice. **g** Hematoxylin-eosin staining of the indicated tissue sections from 8-month-old $Senp3^{+/+}Foxp3$-Cre and $Senp3^{fl/fl}Foxp3$-Cre mice, showing immune cell infiltrations into the tissues of $Senp3^{fl/fl}Foxp3$-Cre mice (arrows). Bars, 100 μm. Data are representative of three or more independent experiments. Error bars are the mean ± SEM values. $n = 5$. Two-tailed unpaired Student's $t$ tests were performed. n.s. not statistically significant; *$P < 0.05$; **$P < 0.01$

antitumor responses in a B16-F10 melanoma model. Compared to the $Senp3^{+/+}Foxp3$-Cre mice, the $Senp3^{fl/fl}Foxp3$-Cre mice displayed a profound reduction in tumor size and in the frequency of tumor-infiltrating Treg cells (Fig. 3e, g). In contrast, the $Senp3^{fl/fl}Foxp3$-Cre mice had an increased frequency of IFN-γ-producing CD4+ and CD8+ Teff cells infiltrating into the tumors (Fig. 3f, g). Parallel studies confirmed that SENP3 ablation in Treg cells significantly suppressed tumor growth and enhanced antitumor immunity in the MC38 colon carcinoma model (Fig. 3h–j). Thus, targeting SENP3 in Treg cells may be an approach to promote antitumor T cell responses.

**SENP3 regulates Treg cell effector programs and stability**. To elucidate the molecular mechanism by which SENP3 regulates the function of Treg cells, we performed RNA sequencing using $Senp3^{+/+}Foxp3$-Cre and $Senp3^{fl/fl}Foxp3$-Cre Treg cells stimulated with anti-CD3 and anti-CD28 for 24 h. Compared to $Senp3^{+/+}Foxp3$-Cre Treg cells, expression of 73 and 82 probes were respectively upregulated and downregulated (fold change > 2 and adjusted $p$ value < 0.01) in $Senp3^{fl/fl}Foxp3$-Cre Treg cells (Fig. 4a, b). To identify key networks regulated by SENP3 in activated Treg cells, we did gene-set enrichment analysis (GSEA)[30]. The analysis of differentially expressed Treg and Teff cell-specific genes revealed that the SENP3-deficient Treg cells express genes associated with Teff cell differentiation, such as $Ifng$, $Il4$, $Il13$, $Il17a$, $Il22$, and $Il9$ (Fig. 4c, d). In contrast, loss of SENP3 in the Treg cells impaired the transcription of Treg cell-specific genes,

such as $Foxp3$ and $Pdcd1$ (Fig. 4c, e). We further validated the expression of the representative genes by qRT-PCR, and $Ifng$, $Il4$, $Il13$, $Il17a$, and $Il21$ were found to be upregulated in the SENP3-deficient Treg cells (Fig. 4f). Further, in an in vitro system to examine Treg cell stability, SENP3-deficient Treg cells showed much lower Foxp3 (Fig. 4g) and exhibited aberrant secretion of IFN-γ and IL-17 (Fig. 4h). These data demonstrated that SENP3 regulates Treg cell effector programs and lineage stability.

**SENP3 catalyzes the deSUMOylation of BACH2**. Previous studies have indicated that various transcription factors control Treg/Teff cell-specific transcriptional programs to maintain the stability and function of Treg cells[7,8]. Because SENP3 localizes to the nucleus to interact with nuclear proteins[18], we sought to determine whether SENP3 mediates deSUMOylation to regulate the activity of Treg cell-specific transcription factors. BACH2 has been shown to maintain the stability and function of Treg cells by repressing Teff cell-specific transcription programs[8]. Interestingly, the transient transfection of HEK293T cells with the respective constructs demonstrated the interaction between SENP3 and BACH2 (Fig. 5a). Additionally, SENP3 was found to physically interact with BACH2 in Treg cells (Fig. 5b). Therefore, we asked whether SENP3 deficiency in Treg cells affects the activity of BACH2. Because BACH2 is generally believed to function through directly binding to the regulatory regions of its target genes, we measured the binding of BACH2 to the regulatory regions of the T helper type (Th) 2 cytokine and $Ccr4$ loci

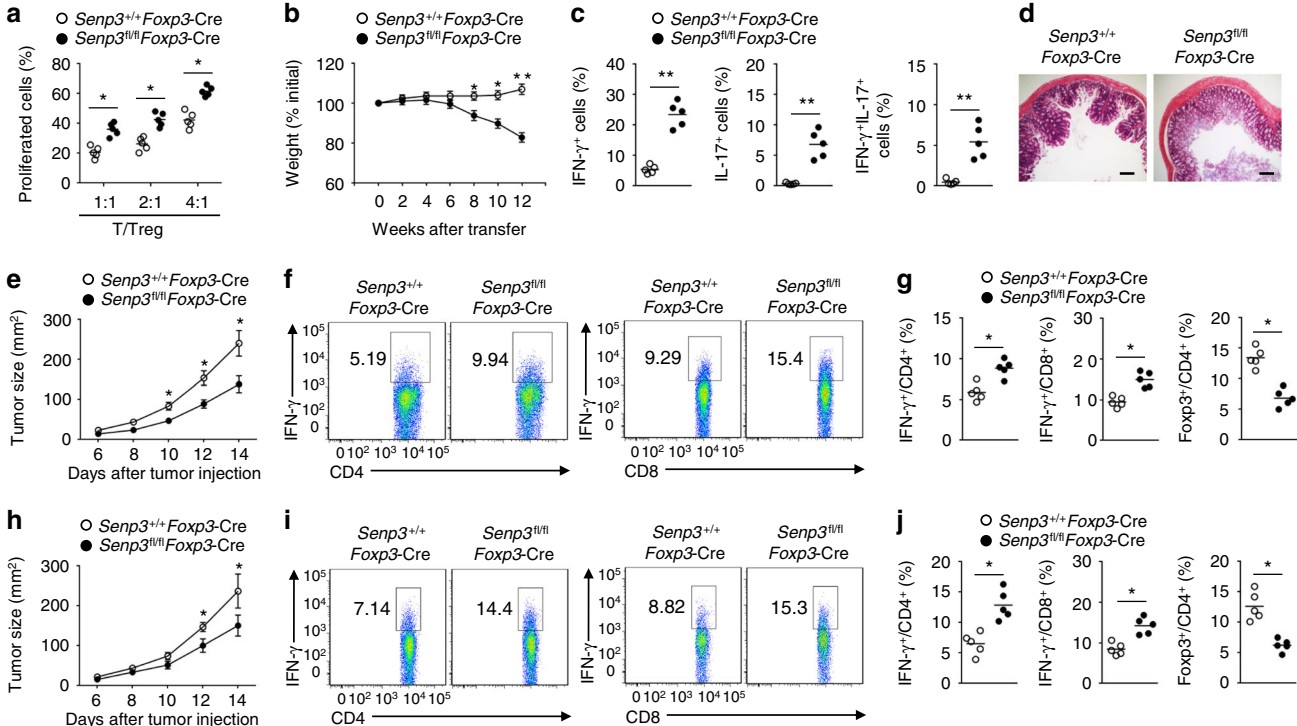

**Fig. 3** SENP3 is required for the suppressive function of Treg cells. **a** CFSE-labeled wild-type CD4+CD25−YFP− T cells were cultured with the indicated number of Senp3+/+Foxp3-Cre or Senp3fl/flFoxp3-Cre Treg cells and stimulated with anti-CD3 and anti-CD28 antibodies for three days. Summary graphs of the percentage of proliferating CFSE-labeled T cells. **b–d** Disease phenotype of 6-week-old RAG-1-deficient mice given adoptive transfer of wild-type naive CD45RBhi CD4+ T cells together with Senp3+/+Foxp3-Cre or Senp3fl/flFoxp3-Cre Treg cells. Body weight was presented relative to initial weight (**b**). Frequency of cytokine-producing CD4+ T cells in the mesenteric lymph nodes (**c**) or hematoxylin-eosin staining of colon sections (**d**) from recipient mice at 12 weeks after adoptive transfer. Bars, 500 μm. **e** Tumor growth in Senp3+/+Foxp3-Cre or Senp3fl/flFoxp3-Cre mice subcutaneously (s.c.) injected with B16-F10 melanoma cells (n = 8). **f, g** Flow cytometric analysis of the frequency of IFN-γ-producing CD4+ and CD8+ T cells or Foxp3+CD4+ T cells in tumors of Senp3+/+Foxp3-Cre and Senp3fl/flFoxp3-Cre mice (day 14 after B16-F10 tumor injection). **h** Tumor growth in Senp3+/+Foxp3-Cre or Senp3fl/flFoxp3-Cre mice subcutaneously (s.c.) injected with MC38 colon cancer cells (n = 8). **i, j** Flow cytometric analysis of the frequency of IFN-γ-producing CD4+ and CD8+ T cells or Foxp3+CD4+ T cells in tumors of Senp3+/+Foxp3-Cre and Senp3fl/flFoxp3-Cre mice s.c. injected with MC38 colon cancer cells (day 14). Data are representative of at least three independent experiments. Error bars are the mean ± SEM values. n = 5 or 8. Two-tailed unpaired Student's t tests were performed. *P < 0.05; **P < 0.01

in Treg cells, which have been reported to be occupied by BACH2 in Th2 cells or Treg cells[8,31]. Compared with that in the wild-type Treg cells, the binding of BACH2 to the regulatory regions of the Teff cell-specific gene loci in SENP3-deficient Treg cells was significantly decreased (Fig. 5c). Thus, we investigated whether SENP3 affects the SUMOylation of BACH2 to mediate its activity. Indeed, BACH2 was conjugated to SUMO3 in the presence of UBC9 and SUMO3 (Fig. 5d). When we expressed BACH2 together with WT SENP3, but not the catalytically inactive (M) SENP3 variant, the SUMOylation of BACH2 was dramatically inhibited (Fig. 5e). Consistent with this, BACH2 was transiently deSUMOylated in wild-type Treg cells upon TCR and CD28 stimulation (Fig. 5f), but this effect was not observed in the SENP3-deficient Treg cells (Fig. 5f), indicating that SENP3 prevents the SUMOylation of BACH2 in activated Treg cells.

Next, using the software of SUMOplot, we predicted SUMOylation motifs on BACH2 at residues K275, K381, K579, and K613. Then, we constructed plasmids expressing WT or mutant BACH2, in which the predicted Lys (K) residues were replaced by Arg (R) to block SUMOylation. BACH2 was conjugated by SUMO3 at the residues K275 and K579 (Fig. 5g). Interestingly, different species harbor two conserved SUMOylation motifs on BACH2 at sites K275 and K579 (Fig. 5h). Consequently, the SUMOylation of K275R and K579R double mutant form of BACH2 (2KR) was nearly abolished (Fig. 5i).

Therefore, K275 and K579 are the main SUMO2/3 conjugation sites on BACH2.

## DeSUMOylation of BACH2 prevents its nuclear export.
SUMOylation mediates protein stability, subcellular localization and protein–protein interaction to regulate the activity of transcription factors[12]. Thus, we first tested whether deSUMOylation affects BACH2 protein stability. Upon TCR and CD28 stimulation, WT and SENP3-deficient Treg cells exhibited similar levels of BACH2 protein (Fig. 6a), suggesting a dispensable role of SENP3 in the regulation of BACH2 stability. Intriguingly, subcellular fractionation assays revealed that the TCR-induced and CD28-induced nuclear localization of BACH2 was inhibited in activated SENP3-deficient Treg cells (Fig. 6a). To examine the function of deSUMOylation in mediating the subcellular localization of BACH2, we transduced naive BACH2-deficient T cells with WT or mutant BACH2 and subsequently cultured these cells under Treg cell polarizing conditions. Compared to the WT or other unrelated mutant BACH2-expressing Treg cells, the K275R or K579R BACH2-expressing Treg cells exhibited increased nuclear levels of BACH2 (Fig. 6b). Particularly, the 2KR BACH2 expressed in Treg cells failed to localize to the cytoplasm (Fig. 6b), suggesting that deSUMOylation facilitates the nuclear localization of BACH2. Since SENP3 localizes to the nucleus to regulate related nuclear proteins, we speculated

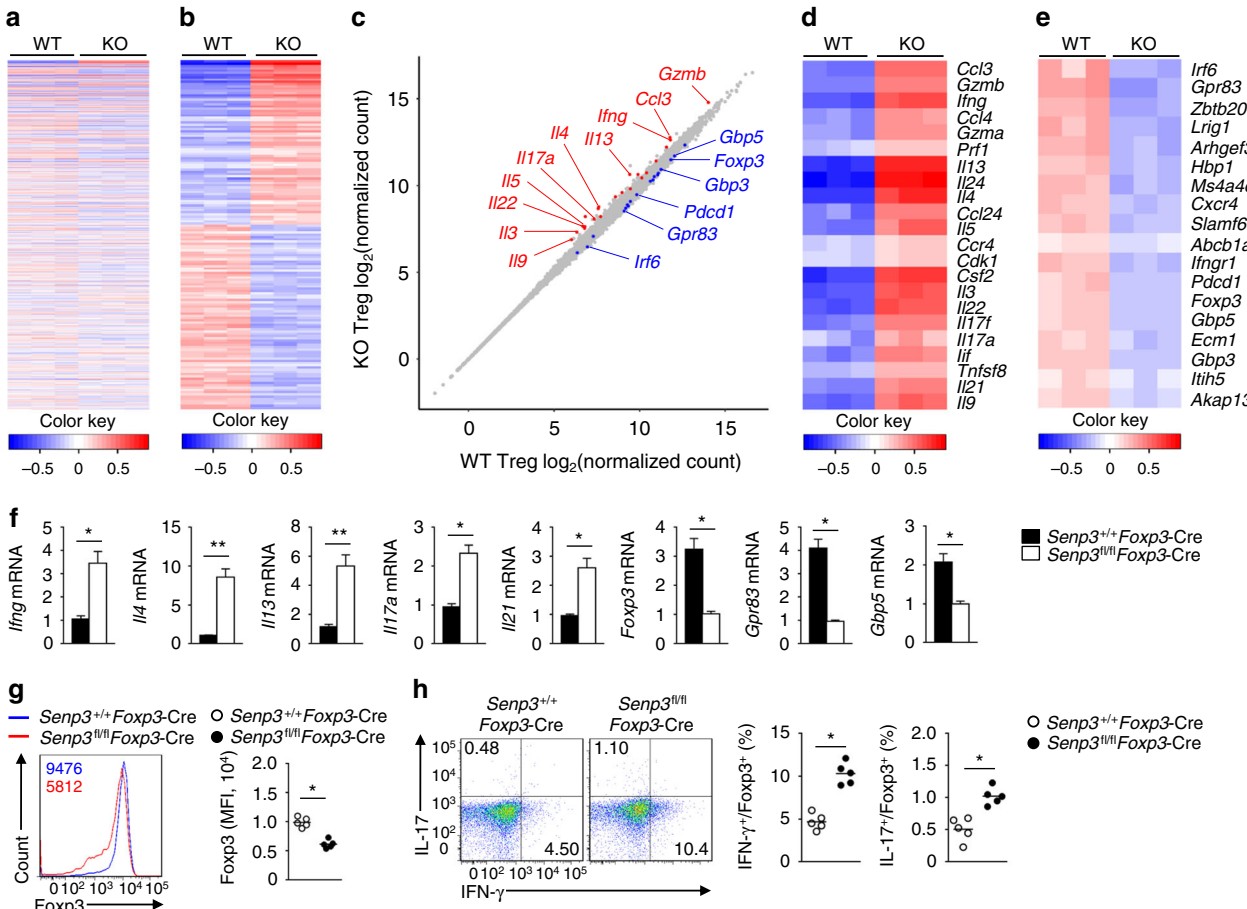

**Fig. 4** SENP3 regulates Treg cell effector programs and stability. **a–e** Splenic Treg cells (CD4+CD25+YFP+) obtained from 6-week-old female *Senp3*+/+*Foxp3*-Cre (WT) mice and *Senp3*fl/fl*Foxp3*-Cre (KO) were stimulated with anti-CD3 and anti-CD28 antibodies for 24 h and subjected to RNA sequencing. A heat map of highly variable genes (top 1000) (**a**) and a heat map of genes with a fold change >2 and adjusted *P* value < 0.01 (**b**) were shown. Scatter plot showing gene expression in the KO versus WT Treg cells (**c**). Teff cell-specific genes are highlighted in blue, and the Treg cell-specific genes are highlighted in red. Clustering of upregulated Teff cell-specific genes identified in SENP3-deficient Treg cells relative to that in *Senp3*-WT Treg cells (**d**). Treg cell-specific gene expression in SENP3-deficient Treg cells and *Senp3*-WT Treg cells (**e**). **f** qRT-PCR analysis of the indicated genes in anti-CD3 and anti-CD28 antibody-stimulated Treg cells from *Senp3*+/+*Foxp3*-Cre and *Senp3*fl/fl*Foxp3*-Cre mice. **g, h** Flow cytometric analysis of Foxp3 expression (**g**, the numbers above the graphs indicate MFI of Foxp3) or IFN-γ and IL-17 expression (**h**) in *Senp3*+/+*Foxp3*-Cre and *Senp3*fl/fl*Foxp3*-Cre Treg cells (CD4+CD25+YFP+) stimulated with anti-CD3 and anti-CD28 antibodies for 24 h. Data are presented as representative plots (left) and summary graphs of percentage and MFI of Foxp3 (right). Data in **f, g, h** are representative of three independent experiments and are presented as the means ± SEM. *n* = 5. Two-tailed unpaired Student's *t* tests were performed. *\*P* < 0.05; *\*\*P* < 0.01

that SENP3 triggers BACH2 deSUMOylation to prevent the nuclear export of BACH2. In WT or mutant BACH2-expressing Treg cells treated with leptomycin B, an inhibitor of nuclear export, BACH2 was mainly localized in the nucleus (Fig. 6c), indicating that the deSUMOylation of BACH2 controls its nuclear export. Consistent with this, in response to treatment with leptomycin B, activated WT and SENP3-deficient Treg cells exhibited similar nuclear levels of BACH2 (Fig. 6d).

To investigate the role of BACH2 deSUMOylation in Treg cell stability, we next isolated Treg cells (CD4+CD25+GFP+) from *Rag1*−/− mice reconstituted with *Bach2*+/+*Cd4*-Cre or *Bach2*fl/fl*Cd4*-Cre bone marrow cells transduced with empty vector (EV) or vector expressing WT or 2KR BACH2 for the in vitro Treg stability assay. Overexpression of WT BACH2 restored the expression of Foxp3 in BACH2-deficient Treg cells (Fig. 6e). Importantly, compared to the WT BACH2-reconstituted BACH2-deficient Treg cells, the 2KR BACH2-reconstituted BACH2-deficient Treg cells exhibited a markedly higher level of Foxp3 (Fig. 6e). Accordingly, the defective expression of Foxp3 in SENP3-deficient Treg cells was reversed by the overexpression of

WT BACH2 (Fig. 6f). Moreover, the 2KR BACH2-reconstituted SENP3-deficient Treg cells produced higher level of Foxp3 than the SENP3-deficient Treg cells reconstituted with WT BACH2 (Fig. 6f). Therefore, BACH2 deSUMOylation prevents its nuclear export to maintain Treg cell stability.

**ROS-induced SENP3 accumulation regulates Treg cell stability.** Because intracellular ROS are elevated during T cell activation[32] and because high level of ROS induces SENP3 stabilization[18], we investigated whether ROS-mediated SENP3 accumulation triggers BACH2 deSUMOylation to maintain Treg cell identity. Interestingly, TCR and CD28 stimulation of Treg cells induced ROS production, leading to the accumulation of SENP3 (Fig. 7a, b). To detect whether the TCR-induced and CD28-induced ROS contributed to SENP3 accumulation in Treg cells, we used N-acetylcysteine (NAC), a widely used antioxidant. Treatment with NAC efficiently restored SENP3 degradation (Fig. 7c). Moreover, the pretreatment of Treg cells with NAC resulted in diminished maintenance of FOXP3 expression (Fig. 7d). Furthermore,

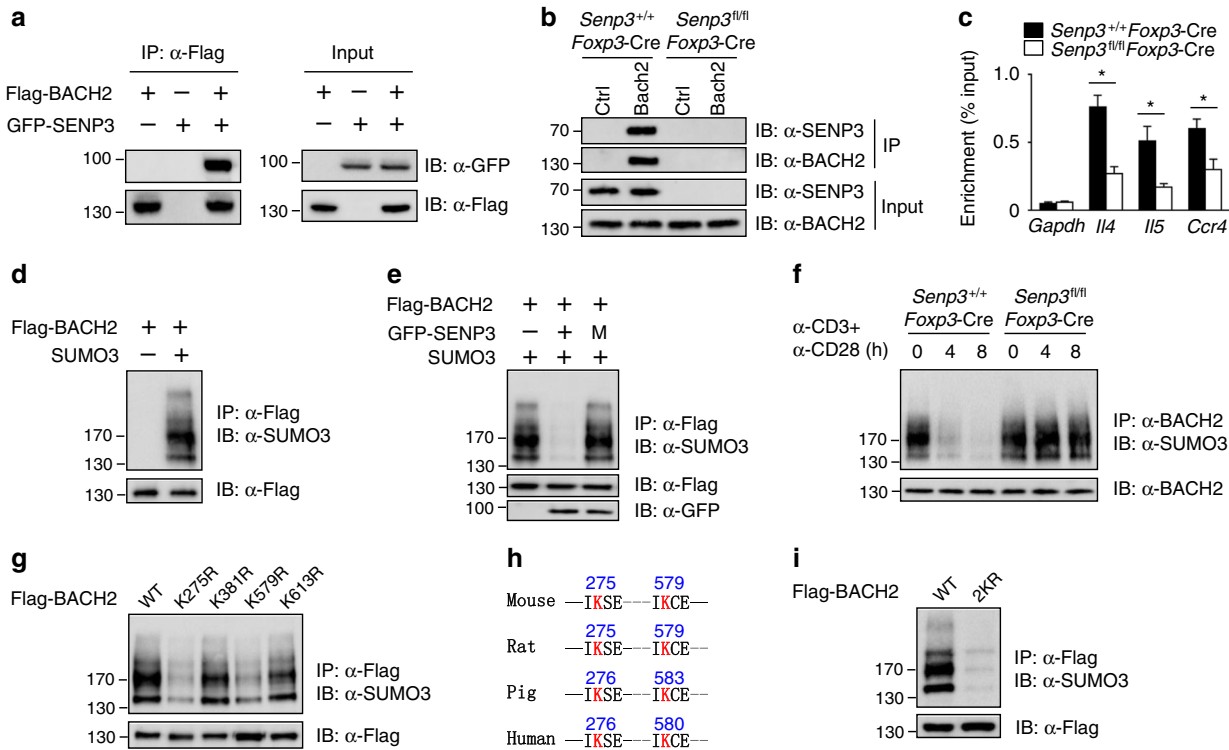

**Fig. 5** SENP3 mediates BACH2 deSUMOylation in Treg cells. **a** SENP3-BACH2 co-immunoprecipitation (co-IP) assays using HEK293 cells transfected with the indicated expression vectors. **b** Lysates from *Senp3*⁺/⁺*Foxp3*-Cre and *Senp3*^fl/fl^*Foxp3*-Cre splenic Treg cells stimulated with anti-CD3 and anti-CD28 antibodies for 24 h were subjected to IP using anti-BACH2 antibody or control Ig (Ctrl); BACH2 and BACH2-associated SENP3 were detected by immunoblotting (IB). **c** ChIP assay to evaluate the binding of BACH2 to the regulatory regions of *Il4*, *Il5*, and *Ccr4* loci in *Senp3*^fl/fl^*Foxp3*-Cre and *Senp3*^+/+^*Foxp3*-Cre splenic Treg cells stimulated with anti-CD3 and anti-CD28 antibodies for 24 h. **d**, **e** BACH2 SUMOylation assays using HEK293 cells transfected with the indicated expression vectors. **f** BACH2 SUMOylation assays using *Senp3*⁺/⁺*Foxp3*-Cre and *Senp3*^fl/fl^*Foxp3*-Cre Treg cells stimulated with anti-CD3 and anti-CD28 antibodies for 0, 4 or 8 h. **g** Flag-tagged mouse BACH2 or its mutant variants were individually transfected into HEK293T cells for BACH2 SUMOylation assays. **h** Sequence alignment of the K275 and K579 SUMOylation sites of BACH2 in different species and conserved residues (red) are shown. **i** Flag-tagged mouse BACH2 (WT) or K275R and K579R mutant BACH2 (2KR) were individually transfected into HEK293T cells for BACH2 SUMOylation assays. All data shown are representative of three independent experiments. Error bars are the mean ± SEM values (**c**). Two-tailed unpaired Student's *t* tests were performed (**c**). *$P < 0.05$

deSUMOylation of BACH2 was prevented in activated Treg cells upon NAC treatment (Fig. 7e), indicating that TCR-stimulated and CD28-stimulated ROS are required for SENP3-mediated BACH2 deSUMOylation and Treg cell stability.

Elevation of ROS can promote tumor-induced immunosuppression[22,23]. We observed that tumor-infiltrating Treg cells exhibited higher level of ROS and SENP3 than did the splenic Treg cells in wild-type mice (Supplementary Fig. 6a, b). To further assess whether SENP3 is involved the regulation of Treg cell-mediated tumor immunosuppression by ROS, we carried out a side-by-side comparison using tumor-bearing *Senp3*⁺/⁺*Foxp3*-Cre, *Senp3*^fl/fl^*Foxp3*-Cre, and *Rag1*⁻/⁻ mice treated with NAC. Indeed, tumor-bearing *Senp3*⁺/⁺*Foxp3*-Cre mice treated with NAC displayed reduced tumor size (Supplementary Fig. 6c). Whereas tumor-bearing *Rag1*⁻/⁻ mice treated with NAC showed similar tumor size as DMSO-treated control mice, indicating an indispensable role of adaptive immunity in the antitumor effect of NAC (Supplementary Fig. 6c). We found that NAC treatment significantly decreased the frequency of tumor-infiltrating Treg cells in *Senp3*⁺/⁺*Foxp3*-Cre mice (Fig. 7f). Furthermore, NAC-treated tumor-infiltrating wild-type Treg cells exhibited increased IFN-γ production (Fig. 7g, h), as well as SUMOylation of BACH2 (Supplementary Fig. 6d). Accordingly, NAC-treated tumor-infiltrating wild-type Treg cells inhibited the proliferation of native T cells in vitro less potently than DMSO-treated tumor-infiltrating wild-type Treg cells (Supplementary Fig. 6e).

Although NAC-treated tumor-infiltrating *Senp3*^fl/fl^*Foxp3*-Cre mice displayed reduced tumor size at day 18 after tumor injection, NAC treatment did not significantly influence tumor size at day 12, 14, and 16 after tumor injection (Supplementary Fig. 6c). In addition, the percentage (Fig. 7f), IFN-γ production (Fig. 7g, h), BACH2 SUMOylation (Supplementary Fig. 6d) or suppressive function (Supplementary Fig. 6e) of tumor-infiltrating Treg cells from *Foxp3*-Cre*Senp3*^fl/f^ mice treated with or without NAC displayed no apparent difference. These data suggested that NAC perturbs tumor-infiltrating Treg cell stability and function via an SENP3/BACH2 deSUMOylation axis.

To confirm the effect of NAC on the stability and function of Treg cells, we injected Treg cells from *Senp3*⁺/⁺*Foxp3*-Cre and *Senp3*^fl/fl^*Foxp3*-Cre (CD45.2⁺) mice pretreated with DMSO or NAC into tumor-bearing B6.SJL (CD45.1⁺) mice using CD45.1 and CD45.2 markers to clearly separate the host effect of T effector cells and donor Treg cells. Compared to the NAC-treated wild-type Treg cells, the untreated wild-type Treg cells were significantly more potent at promoting tumor growth of B6.SJL mice (Fig. 7i). In contrast, the B6.SJL mice injected with SENP3-deficient Treg cells treated with or without NAC displayed no apparent difference in tumor growth (Fig. 7i). We observed that NAC treatment increased the percentage of tumor-infiltrating wild-type CD45.2⁺ Foxp3⁻ T cells, enhanced the expression of IFN-γ in tumor-infiltrating wild-type Treg cells and promoted the tumor infiltration of CD8⁺ effector T cells (Fig. 7j–m). Of note,

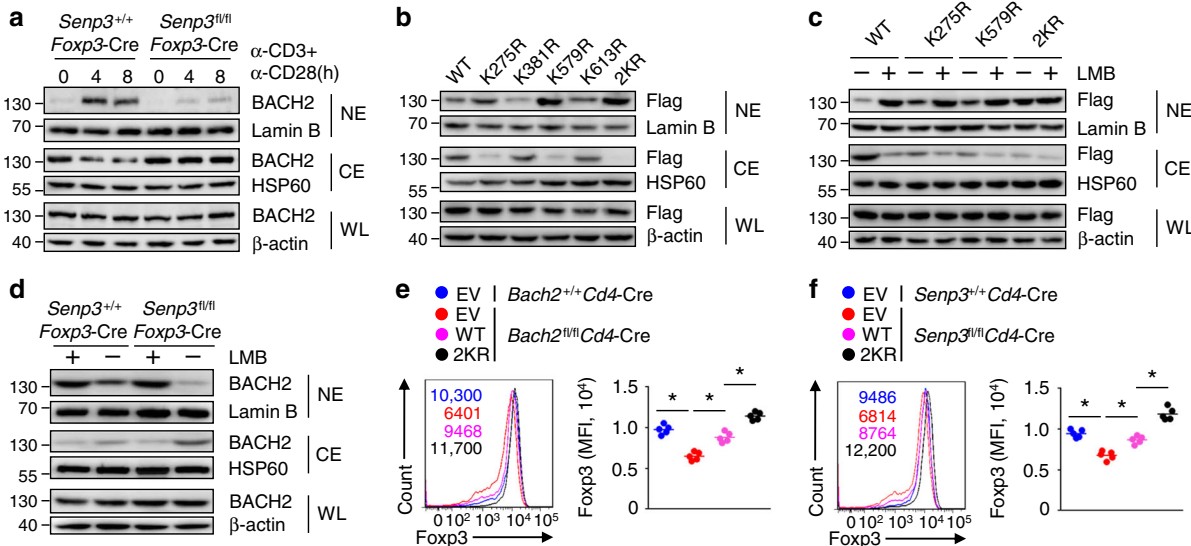

**Fig. 6** BACH2 deSUMOylation regulates its nuclear export in Treg cells. **a** IB analysis of the indicated proteins in whole (WL), nuclear (NE), and cytoplasmic (CE) extracts of *Senp3*^+/+*Foxp3*-Cre and *Senp3*^fl/fl*Foxp3*-Cre Treg cells stimulated with anti-CD3 and anti-CD28 antibodies for 0, 4 or 8 h. **b** IB analysis of the indicated proteins in anti-CD3 and anti-CD28 antibody-stimulated naive CD4^+ T cells from *Bach2*^fl/fl*Cd4*-Cre mice transduced with the indicated lentiviral vector encoding Flag-WT BACH2 or Flag-mutant BACH2 under Treg cell conditions. **c** IB analysis of the indicated proteins in different lentivirus-transduced BACH2-deficient CD4^+ T cells under Treg cell conditions treated with 10 ng/ml of leptomycin B for 4 h. **d** IB analysis of *Senp3*^+/+*Foxp3*-Cre and *Senp3*^fl/fl*Foxp3*-Cre splenic Treg cells stimulated with anti-CD3 and anti-CD28 antibodies in the presence of leptomycin B for 4 h. **e** Splenic Treg cells (CD4^+CD25^+GFP^+) from *Rag1*^−/− mice reconstituted (for 8 weeks) with *Bach2*^+/+*Cd4*-Cre or *Bach2*^fl/fl*Cd4*-Cre bone marrow (BM) cells transduced with empty vector (EV) or vector expressing WT or 2KR BACH2 were stimulated with anti-CD3 and anti-CD28 antibodies for 24 h and subjected to flow cytometric analysis of Foxp3 expression. **f** Splenic Treg cells from *Rag1*^−/− mice reconstituted with *Senp3*^+/+*Cd4*-Cre or *Senp3*^fl/fl*Cd4*-Cre BM cells transduced with EV, WT or 2KR BACH2 were stimulated with anti-CD3 and anti-CD28 antibodies for 24 h and subjected to flow cytometric analysis of Foxp3 expression. Numbers above graphs indicate MFI of Foxp3 (**e**, **f**). Data shown are representative of three independent experiments. Error bars are the mean ± SEM values (**e**, **f**). Two-tailed unpaired Student's *t* tests were performed (**e**, **f**). *$P < 0.05$

NAC treatment did not significantly affect the percentage of SENP3-deficient CD45.2^+ Foxp3^− T cells, IFN-γ expression and suppressive function of SENP3-deficient Treg cells in tumors (Fig. 7j–m), indicating that treatment with NAC perturbs tumor-infiltrating Treg cell stability and function in an SENP3-dependent manner, thereby contributing to the antitumor effect of NAC. Therefore, the ROS-dependent accumulation of SENP3 following TCR and CD28 stimulation maintains Treg cell stability and function.

## Discussion

We identified SENP3 as a positive regulator of Treg cell stability and function. SENP3 deficiency resulted in dysregulated T cell homeostasis, spontaneous autoimmune symptoms and enhanced antitumor immunity. SENP3-triggered BACH2 deSUMOylation in Treg cells regulated the nuclear localization and transcriptional activity of BACH2 to repress Teff cell-specific transcriptional programs and maintain Treg cell-specific gene signatures. Importantly, SENP3 was rapidly stabilized by TCR-stimulated and CD28-stimulated ROS, resulting in BACH2 deSUMOylation to maintain Treg cell stability and function (Supplementary Fig. 7). Our work provides deep insights into the regulation and function of SENP3-mediated deSUMOylation in autoimmunity and antitumor immunity.

SUMOylation has been reported to be required for Treg cell expansion and function, but whether deSUMOylation is required for Treg cell activity remains elusive. We observed that the SENP3 deficiency compromised the expression of Treg cell-specific genes and impaired the suppressive activity of Treg cells. Our study unveils the role of deSUMOylation in Treg cell-mediated T cell homeostasis and immune tolerance. Although the

deconjugation of SUMO1 by the ectopic expression of SENP1 inhibits T cell activation[15], SENP3 is dispensable for T cell activation. Our data suggest that the regulation of T cell activation may be dependent on the dynamic deconjugation of SUMO1 or SUMO2/3. Indeed, it is still unclear whether SENP3 regulates Th1, Th2, Th17 or CD8^+ T cell differentiation. Therefore, the functional importance of SENP3 in different types of T cells and disease models remains to be further studied.

BACH2 is a well-known transcriptional repressor involved in the development and function of a diversity of innate and adaptive immune cells[33–37]. Particularly, BACH2 promotes the differentiation of Treg cells, which is required to establish immune tolerance and homeostasis[8,9,38,39]. An important question is whether the post-translational modification of BACH2, in particular SUMOylation, regulates its activity. Our data revealed a deSUMOylation-dependent function of BACH2 in the regulation of Treg cell-mediated immunosuppression. In resting Treg cells, BACH2 is predominantly conjugated to SUMO2/3 and shuttled between the cytoplasm and nucleus; however, the TCR-induced and CD28-induced ROS-mediated SENP3 accumulation triggered the deSUMOylation of BACH2, which promoted the nuclear localization and transcriptional activity of BACH2. Nevertheless, it remains to be answered why BACH2 needs both K275 and K579 SUMOylation sites. Because SUMOylation on either site can promote the nuclear export of BACH2 in Treg cells, as determined by the fractionation of cellular lysates after treatment with leptomycin B, having two sites should increase the SUMOylated fraction of BACH2, resulting in more BACH2 molecules to be exported from the nucleus. Future studies are thus needed to examine the potential role of K275 and K579 SUMOylation sites in the regulation of BACH2 activity in different types of immune cells.

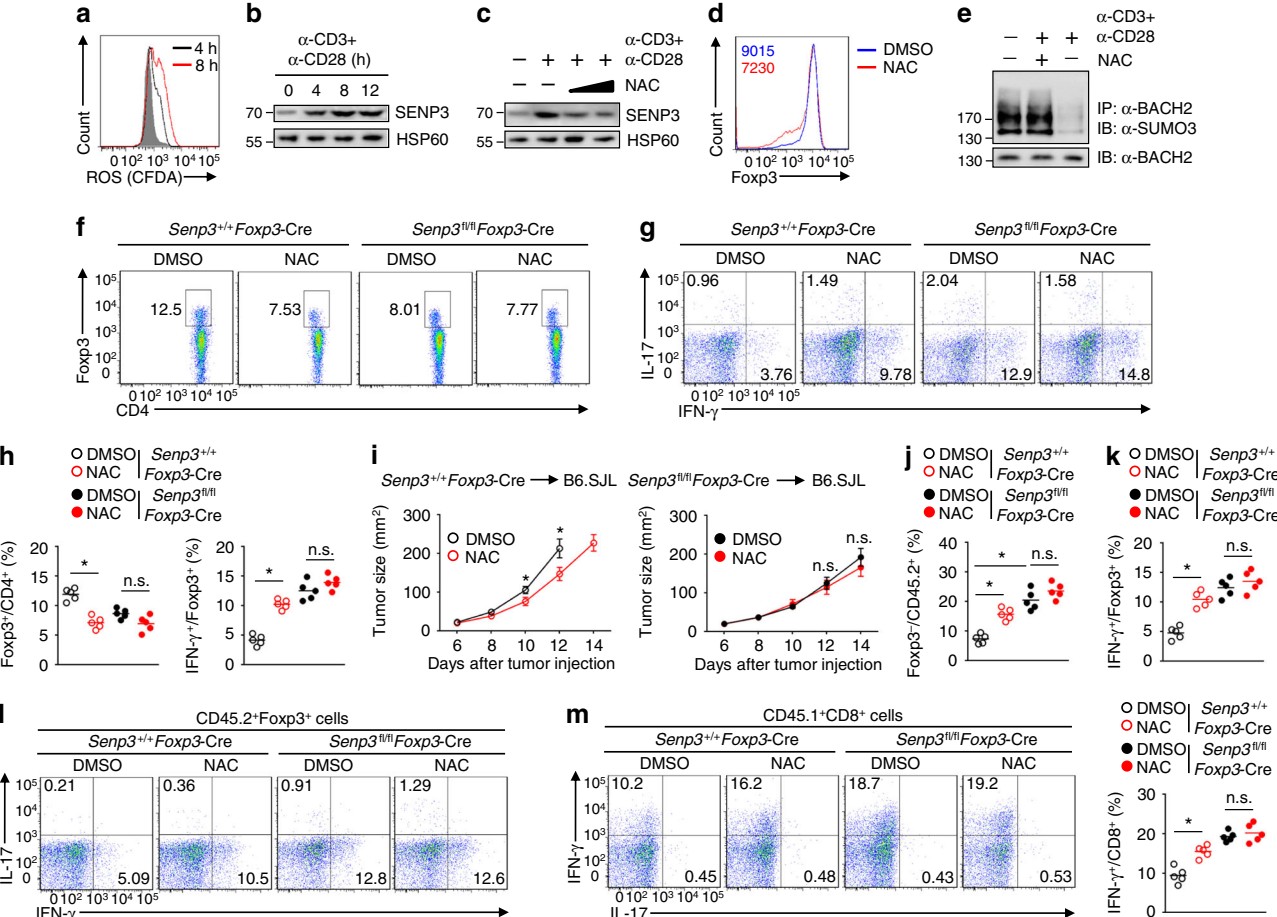

**Fig. 7** ROS-induced SENP3 accumulation regulates Treg cell stability. **a, b** Levels of ROS (**a**) and SENP3 (**b**) in Treg cells stimulated with anti-CD3 and anti-CD28 antibodies for the indicated time periods. Grey area, unstimulated Treg cells. CFDA, CM-H2DCFDA. **c** SENP3 expression in Treg cells stimulated or not with anti-CD3 and anti-CD28 antibodies for 8 h in the presence or absence of 0.5 mM or 5 mM NAC. **d** Foxp3 expression in Treg cells stimulated with anti-CD3 and anti-CD28 antibodies for 12 h in the presence or absence of 5 mM NAC. **e** SUMOylation assays using Treg cells stimulated or not with anti-CD3 and anti-CD28 for 8 h in the presence or absence of 5 mM NAC. **f–h** Flow cytometric analysis of percentage (**f**) or IFN-γ and IL-17 expression (**g, h**) of tumor-infiltrating Treg cells from tumor-bearing $Senp3^{+/+}Foxp3$-Cre and $Senp3^{fl/fl}Foxp3$-Cre mice receiving daily NAC (on day 14). **i–m** B6.SJL mice were injected (s.c.) with B16-F10 cells and then intravenously (i.v.) treated (on day 6) with $Senp3^{+/+}Foxp3$-Cre and $Senp3^{fl/fl}Foxp3$-Cre Treg cells ($2 \times 10^6$ cells per mouse) stimulated with anti-CD3 and anti-CD28 antibodies for 12 h in the presence or absence of 5 mM NAC. **i** Growth curves of tumors in B6.SJL mice ($n = 8$ mice per group). **j** Percentage of tumor-infiltrating $Foxp3^-CD45.2^+$ T cells (on day 14). **k–m** Flow cytometric analysis of IFN-γ and IL-17 expression in tumor-infiltrating $CD45.2^+Foxp3^+$ (**k, l**) or $CD45.1^+CD8^+$ (**m**) T cells (on day 14). Data are representative of three independent experiments and are presented as the mean ± SEM. $n = 5$ or 8. Two-tailed unpaired Student's $t$ tests were performed. n.s. not statistically significant; *$P < 0.05$

Previous studies have suggested that excessive ROS levels are associated with tumor-induced immunosuppression[22,23] and that ROS can participate in Treg cell-mediated immunosuppression[24]. However, the mechanism by which ROS affect Treg cell function remains poorly defined. Although the ROS scavenger NAC is a promising cancer chemo-preventive agent that acts through a variety of mechanisms, including its nucleophilicity, antioxidant activity, modulation of metabolism, regulation of cell survival, and apoptosis, influence on DNA repair and anti-inflammatory activity[40], the effect of NAC on Treg cell-mediated tumor immunosuppression in vivo has not yet been elucidated. In this study, we found SENP3 was rapidly stabilized by TCR-stimulated and CD28-stimulated ROS, resulting in BACH2 deSUMOylation to maintain the stability and function of Treg cells. Our results further indicate that the antitumor effect of NAC is contributed, at least in part, by NAC-induced impaired Treg cell stability and function via an SENP3/BACH2 deSUMOylation axis. Our current study describes a molecular mechanism underlying the cross-talk between ROS and Treg cell-mediated tumor immunosuppression. These findings suggest that targeting ROS in Treg cells may be an effective approach to ameliorate SENP3-mediated tumor immune tolerance.

In summary, our data reveal that SENP3 maintains the stability and function of Treg cells via BACH2 deSUMOylation and thereby regulates T cell homeostasis and immune tolerance. Our results identify the role of SENP3-triggered BACH2 deSUMOylation in the cross-talk between ROS and Treg cell-mediated tumor immunosuppression. Our findings also provide a promising strategy to enhance antitumor immunity and improve T cell-based immunotherapy via targeting SENP3.

## Methods

**Mice.** *Senp3*-floxed mice (in C57BL/6 background) were generated at the Model Animal Research Center of Nanjing University (Nanjing, China) using a LoxP targeting system. The *Senp3*-floxed mice were crossed with *Cd4*-Cre transgenic mice (The Jackson Laboratory) in B6 background to produce age-matched *Senp3*$^{+/+}$*Cd4*-Cre and *Senp3*$^{fl/fl}$*Cd4*-Cre mice for experiments. The *Senp3*-floxed mice were crossed with *Foxp3*-YFP-Cre transgenic mice (The Jackson Laboratory) in B6 background to produce age-matched *Senp3*$^{+/+}$*Foxp3*-Cre and *Senp3*$^{fl/fl}$*Foxp3*-Cre mice for experiments. *Bach2*-floxed mice (in C57BL/6 background) were generated at the Wellcome Trust Sanger Institute using a LoxP targeting system. The *Bach2*-

floxed mice were crossed with *Cd4*-Cre transgenic mice to produce age-matched *Bach2*$^{+/+}$*Cd4*-Cre and *Bach2*$^{fl/fl}$*Cd4*-Cre mice for experiments. B6.SJL mice (expressing the CD45.1 congenic marker), *Rag1*-KO mice and OT-I TCR-transgenic mice in C57BL/6 background were from the Jackson Laboratory. Mice were maintained in a specific pathogen-free (SPF) facility, and all animal experiments were in accordance with protocols approved by the Institutional Animal care and Use Committee (IACUC) of Shanghai Jiao Tong University, School of Medicine.

**Plasmids and reagents**. Flag-tagged mouse BACH2 and mutant BACH2 were cloned into the lentiviral vector pLVX-IRES-ZsGreen1. SENP3 and catalytically inactive (M) SENP3 were cloned into the pEGFP-C1 vector[18,19]. Antibodies for HSP60 (H1, sc-13115), GFP (B-2, sc-9996) and Lamin B (C-20, sc-6216) were from Santa Cruz Biotechnology, Inc. Antibodies for SENP3 (D20A10, #5591S) and SUMO-2/3 (18H8, #4971S) were purchased from Cell Signaling Technology. HRP-conjugated anti-HA antibody (3F10) was from Roche. Anti-β-actin (AC-74, A2228) and anti-Flag (M2, F3165) antibodies were from Sigma-Aldrich. Anti-BACH2 antibody was validated and used in previous studies[41,42]. The fluorochrome-conjugated antibodies for CD4 (GK1.5), CD8 (53–6.7), CD44 (IM7), CD62L (MEL-14), CD69 (H1.2F3), Foxp3 (FJK-16s), IFN-γ (XMG1.2), and IL-17 (eBio17B7) were from eBioscience. Primary antibody dilution of 1:1000 and 1:200 was used for immunoblotting assay and flow cytometric assay, respectively. N-Acetyl-L-cysteine (NAC) and leptomycin B were purchased from Sigma-Aldrich. All the uncropped scans of the western blots are shown in Supplementary Fig. 8.

**Histology**. Organs were removed from *Senp3*$^{+/+}$*Cd4*-Cre and *Senp3*$^{fl/fl}$*Cd4*-Cre mice or *Senp3*$^{+/+}$*Foxp3*-Cre and *Senp3*$^{fl/fl}$*Foxp3*-Cre mice, fixed in 10% neutral buffered formalin, embedded in paraffin, and sectioned for staining with hematoxylin and eosin.

**Flow cytometry**. For analysis of surface markers, cells were stained in PBS containing 2% fetal bovine serum (FBS) with antibodies as indicated. Foxp3 staining was performed according to the manufacturer's instructions (eBioscience). To determine cytokine expression, cells were stimulated with phorbol12-myristate 13-acetate (PMA), ionomycin, monensin for 5 h. At the end of stimulation, cells were stained with the indicated antibodies according to the manufacturer's instructions (eBioscience). For analyzing in vivo primed antigen-specific T cells, cells from tumor tissues were stimulated in vitro with the indicated antigenic peptides in the presence of monensin and then subjected to ICS. ROS were measured by incubation with 10 μM CM-H2DCFDA (5-(and-6)-chloromethyl-2,7-dichlorodihydrofluorescein diacetate acetyl ester; Invitrogen) for 30 min at 37 °C. All FACS gating strategies are shown in Supplementary Fig. 9.

**T cell isolation and stimulation**. Primary T cells were isolated from the spleen and lymph nodes of female mice (6 weeks old). Naive CD4$^+$, CD8$^+$ T cells and Treg cells were purified by flow cytometric cell sorting based on CD4$^+$CD44$^{lo}$CD62L$^{hi}$, CD8$^+$CD44$^{lo}$ and CD4$^+$CD25$^+$YFP$^+$ markers, respectively. The cells were stimulated with plate-bound anti-CD3 (1 μg/ml) and anti-CD28 (1 μg/ml) in replicate wells of 96-well plates (1 × 10$^5$ cells per well) for ELISA using a commercial assay system (eBioScience). Naive CD4$^+$ T cells were stimulated with anti-CD3 (5 μg/ml) plus anti-CD28 (1 μg/ml) under T$_H$1 (5 μg/ml anti-IL4, 10 ng/ml IL-12), T$_H$2 (5 μg/ml anti-IFN-γ, 20 ng/ml IL-4), T$_H$17 (5 μg/ml anti-IL4, 5 μg/ml anti-IFN-γ, 15 ng/ml IL-6, 2.5 ng/ml TGF-β) and Treg (5 μg/ml anti-IL4, 5 μg/ml anti-IFN-γ, 5 ng/ml TGF-β) conditions.

**RNA preparation and qRT-PCR**. Total RNA was extracted using TRIzol reagent (Invitrogen). cDNA was synthesized using a reverse transcriptase kit (TaKaRa, Japan). Real-time qRT-PCR was performed using gene-specific primer sets (Supplementary Table 1). Gene expression was assessed in triplicate and normalized to a reference gene, *Actβ*.

**RNA-sequencing analysis**. Fresh splenic Treg cells (CD4$^+$CD25$^+$YFP$^+$) were isolated from 6-week-old *Senp3*$^{+/+}$*Foxp3*-Cre and *Senp3*$^{fl/fl}$*Foxp3*-Cre mice and stimulated with anti-CD3 and anti-CD28 for 24 h. Activated Treg cells were used for total RNA isolation with TRIzol (Invitrogen) and subjected to RNA-sequencing using Illumina Nextseq500 (75 bp paired end reads). The raw reads were aligned to the mouse reference genome (version mm10), using Tophat2 RNASeq alignment software[43]. The mapping rate was 96% overall across all the samples in the dataset. HTSeq was used to quantify the gene expression counts from Tophat2 alignment files[44]. Differential expression analysis was performed on the count data using R package DESeq2[45]. *P*-values obtained from multiple tests were adjusted using Benjamini-Hochberg correction. Significant differentially expressed genes are defined by a Benjamini-Hochberg corrected *p*-value cutoff of 0.05 and fold-change of at least one.

**Immunoblotting and immunoprecipitation**. Cells were washed with ice-cold PBS and lysed on ice for 30 min in RIPA buffer (50 mM Tris-HCl, pH 7.5; 135 mM NaCl; 1% NP-40; 0.5% sodium DOC; 1 mM EDTA; 10% glycerol) containing protease inhibitor (1:100, P8340; Sigma-Aldrich), 1 mM NaF, and 1 mM PMSF.

Cell lysates were cleared by centrifugation, and supernatants were immunoprecipitated with the appropriate antibodies using protein A/G-agarose beads. Samples were then used for immunoblotting analysis with indicated antibodies.

**ChIP assay**. Treg cells stimulated with plate-bound anti-CD3 and anti-CD28 for 24 h were crosslinked with 1% formaldehyde and neutralized with 0.125 M glycine. Cell lysates were sonicated and proteins were immunoprecipitated with antibody to BACH2 or IgG as a control. After complete washing, immunoprecipitated DNA was eluted in elution buffer and reverse-crosslinked overnight at 65 °C. DNA was purified and quantified by real-time PCR (primer sequences, Supplementary Table 2). Enrichment was calculated relative to input (%).

**Analysis of endogenous BACH2 SUMOylation**. A well-established method was used to detect endogenous BACH2 SUMOylation by immunoprecipitation (IP)[46]. In brief, HEK293 or Treg cell pellets lysed by adding 200 ml of the lysis buffer (62.5 mM Tris pH 6.8, 2%SDS) and boiling for 10 min. The samples were centrifuged for 20 min. The supernatant was transferred to a new tube and diluted 1/20 with NEM-RIPA buffer. IP with anti-Flag or anti-BACH2 antibody was performed and immunoprecipitates were resolved by SDS-PAGE following immunoblot assay with anti-SUMO2/3 antibody.

**Lentiviral transduction and bone marrow reconstitution**. Lentiviral transduction was performed by transfecting HEK 293T cells with pLVX-IRES-ZsGreen1 lentiviral vectors along with packaging plasmids[47]. Packaged pLVX lentivirus was produced and used to infect activated T cells. Purified naive CD4$^+$ T cells (CD4$^+$CD44$^{lo}$CD62L$^{hi}$) were activated with plate-bound anti-CD3 (5 μg/mL) plus anti-CD28 (1 μg/mL) in 48-well plates for 24 h and then infected with the lentivirus in the presence of 10 μg/ml polybrene by spinning at 900 g for 90 min. Transduced cells were used for in vitro Treg cell differentiation for 3 days. Bone marrow cells from *Bach2*$^{+/+}$*Cd4*-Cre, *Bach2*$^{fl/fl}$*Cd4*-Cre, *Senp3*$^{+/+}$*Cd4*-Cre or *Senp3*$^{fl/fl}$*Cd4*-Cre mice were cultured for 24 h in IL-3 (10 ng/ml), IL-6 (10 ng/ml), and SCF (100 ng/ml) containing complete DMEM and then infected with packaged lentivirus for 2 days. Transduced bone marrow cells were injected into lethally irradiated (950 rads) *Rag1*$^{-/-}$ recipient mice. *Rag1*$^{-/-}$ mice were euthanized and analyzed 8 weeks after reconstitution.

**Tumor models**. B16-F10 murine melanoma cells were obtained from the ATCC. B16-OVA (B16-expressing OVA) murine melanoma cells and MC38 murine colon cancer cells were kindly gifted by Shao-Cong Sun, Department of Pathology, The University of Texas MD Anderson Cancer Center. We confirmed that all cell lines were tested negative for mycoplasma contamination. All the cell lines used in this study have been authenticated by previously reported data. These tumor cells were cultured in RPMI 1640 supplemented with 10% FBS and injected s.c. into 6-week-old *Senp3*$^{+/+}$*Foxp3*-Cre and *Senp3*$^{fl/fl}$*Foxp3*-Cre mice (5 × 10$^5$ cells per mouse). The challenged mice were monitored for tumor growth, and the tumor size was expressed as tumor area. To minimize individual variations, age-matched and sex-matched mice used were allocated randomly. For the tumor models, the data were collected by two investigators independently. The investigators were blinded to group allocation during data collection and analysis.

**N-acetylcysteine treatment**. In vitro, NAC was added at 5 mM to T or Treg cells for time points indicated. In vivo, mice receiving daily NAC with drinking water (12.25 mmol/kg body weight) starting 16 h after injection of B16-F10 melanoma cells.

**Statistical analysis**. Statistical analysis was performed using Prism software (GraphPad Prism version 6.01). Two-tailed unpaired Student's *t* tests were performed, and data are presented as means ± SEM. One-way ANOVA, where applicable, was performed to determine whether an overall statistically significant change existed before the Student's *t* test to analyze the difference between any two groups. *P* value < 0.05 is considered statistically significant, and the level of significance was indicated as *$P$ < 0.05 and **$P$ < 0.01.

**Data availability**. The RNA-seq data reported in this paper are available under accession number SRP127517 (NCBI Trace and Short-Read Archive). All other data are available from the authors upon reasonable requests.

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

## Acknowledgements

This study was supported by grants from the National Natural Science Foundation of China (31230037, 31670896, 31370904, 81671579, 91753141), the Ministry of Science and Technology of China (2013CB910900), the National Key Research and Development Program (2017YFA0104500), the Recruitment Program of Global Experts of China, Shanghai Rising-Star Program (16QA1403300), Shanghai Municipal Commission of Health and Family Planning (20174Y0049, 20174Y0191), Shanghai Jiao Tong University "Program for young teachers" (KJ30214170006) and "Medical and Engineering Cross Research Foundation" (YG2016QN77).

## Author contributions

X.Y., Y.L., and X.-L.T. designed and performed the experiments, prepared the figures, and wrote the manuscript; S.L., Y.Z., F.W., X.G., S.D., Y.C., Z.L., L.C., and L.-M.L. contributed to the performance of the experiments, X.W., J.C., B.L., B.S., and J.J. contributed critical comments, H.-B.L., C.H., J.Y., and Q.Z. supervised the work and wrote the manuscript.

## Additional information

**Competing interests:** The authors declare no competing interests.

