## [Peer Review File · Nature Communications]

Reviewers' comments:

Reviewer #1 (Remarks to the Author):

The authors have adequately addressed my previous concerns and the manuscript is significantly improved. I have no more questions.

Reviewer #2 (Remarks to the Author):

In this revision, Dr. Qiang Zou and colleagues produced a large collection of data to improve their manuscript and address the concerns of the reviewers. This reviewer appreciates their efforts and recognizes that substantial progress has been made. However, a few lingering issues still exist that, in this reviewer's opinion, preclude the current version to meet the standard for publication.

1. One of the main issue raised from last round was the distinctions between Treg maintenance and induction. The authors added nTreg stability data in Fig. 4g, which is helpful. However, Fig. 4i is an induced-Treg assay. This reviewer cannot see why this result support "an important role of SENP3 in maintaining the stability of Treg cells" (Line 205). Additionally, the experiments in Fig. 6e and 6f are still iTreg assays, and the authors' conclusion "Therefore, BACH2 deSUMOylation prevents its nuclear export to maintain Treg cell stability." (Line 275) These experiments need to be done with nTregs to support the conclusion, otherwise the conclusion need to be modified to reflect the data.
2. The author added a new allelically marked Treg transfer experiment in Fig. 7j and 7k to show Senp3^{-/-} Tregs are less stable in tumor microenvironment. However, they didn't show the percentage of CD45.2+ transferred Tregs that lost Foxp3 expression in the tumor. This number is the key data to prove the authors' statement on Treg instability.
3. The FACS plots in Fig. 2a do not seem to be correct (it's different from the previous version). For example, 16.2% Foxp3+ gate in spleen doesn't seem to match the actual dot plot shown.

Response from reviewer 2 regarding concerns of reviewer 1 from previous round of review:

Overall, I feel the revised manuscript largely addressed reviewer 1's critiques for the original version of the manuscript. There are 3 main points from reviewer 1.

1. The authors might be overstating the effect of NAC in suppressing tumor growth. In the revised manuscript, the authors added new evidence showing that NAC's effect on tumor is partly dependent on Treg cells and targets SENP3. They also softened the conclusion significantly compared to the original version, which I think it's acceptable now.
2. Some of the RT-qPCR data and FACS cytokine profiles look very similar and seem too good to be true. The authors repeated a lot of RT-qPCR and FACS experiments shown in Fig. 4 (previously Fig. 5). The current results (Fig. 4f and 4h) look very realistic compared to the results shown in the original version. I did notice that a lot of genes were removed from the RT-qPCR and FACS analysis in Fig. 4 this time. They also completely removed RT-PCR result in Fig. 8d. There was no mentioning in their response on why those results were removed. Nevertheless, I think the data shown in the revised manuscript is adequate to support their conclusion.
3. The RNA-seq data analysis is biased. The authors added fig. 4a and 4b, and explained how they selected the subsets of genes that are differentially expressed in WT vs KO Tregs. They also pointed out that GSEA method was used to narrow down the T effector cell and Treg signature genes shown in Fig. 4c to 4e. I do think their approach is reasonable here.

In conclusion, I think the quality of this revision improved quite a bit compared to the original version, which did contain some dubious results. They still have to address the questions I raised last time before I feel comfortable to give my green light.

Reviewer #1:

The authors have adequately addressed my previous concerns and the manuscript is significantly improved. I have no more questions.

Response: We thank the reviewer for her/his careful evaluation of our manuscript.

Reviewer #2:

In this revision, Dr. Qiang Zou and colleagues produced a large collection of data to improve their manuscript and address the concerns of the reviewers. This reviewer appreciates their efforts and recognizes that substantial progress has been made. However, a few lingering issues still exist that, in this reviewer's opinion, preclude the current version to meet the standard for publication.

Response: We thank the reviewer for her/his positive comments on our work. We also thank the reviewer for this insightful comment and have now included the additional data to address the concerns.

1. One of the main issue raised from last round was the distinctions between Treg maintenance and induction. The authors added nTreg stability data in Fig. 4g, which is helpful. However, Fig. 4i is an induced-Treg assay. This reviewer cannot see why this result support "an important role of SENP3 in maintaining the stability of Treg cells" (Line 205). Additionally, the experiments in Fig. 6e and 6f are still iTreg assays, and the authors' conclusion "Therefore, BACH2 deSUMOylation prevents its nuclear export to maintain Treg cell stability." (Line 275) These experiments need to be done with nTregs to support the conclusion, otherwise the conclusion need to be modified to reflect the data.

Response: The reviewer's point is well taken. Splenic nTreg cells (CD4⁺CD25⁺GFP⁺) from *Rag1*^{-/-} mice reconstituted (for 8 weeks) with *Bach2*^{+/+} *Cd4*-Cre or *Bach2*^{fl/fl} *Cd4*-Cre bone marrow cells transduced with empty vector (EV) or vector expressing WT or 2KR BACH2 were isolated and stimulated with anti-CD3 and anti-CD28 antibodies for 24 hours. These stimulated-nTreg cells were then subjected to flow cytometric analysis of Foxp3 expression. Overexpression of WT BACH2 restored the expression of Foxp3 in BACH2-deficient Treg cells (Fig. 6e). Importantly, compared to the WT BACH2-reconstituted BACH2-deficient Treg cells, the 2KR BACH2-reconstituted BACH2-deficient Treg cells exhibited a markedly higher level of

Foxp3 (Fig. 6e). Splenic nTreg cells (CD4⁺CD25⁺GFP⁺) from *Rag1*^{-/-} mice reconstituted (for 8 weeks) with *Senp3*^{+/+}*Cd4*-Cre or *Senp3*^{fl/fl}*Cd4*-Cre bone marrow cells transduced with empty vector (EV) or vector expressing WT or 2KR BACH2 were also stimulated with anti-CD3 and anti-CD28 antibodies for 24 hours and subjected to flow cytometric analysis of Foxp3 expression. The defective expression of Foxp3 in SENP3-deficient Treg cells was reversed by the overexpression of WT BACH2 (Fig. 6f). Moreover, the 2KR BACH2-reconstituted SENP3-deficient Treg cells produced higher level of Foxp3 than the SENP3-deficient Treg cells reconstituted with WT BACH2 (Fig. 6f). These data demonstrated that BACH2 deSUMOylation maintains Treg cell stability. As suggested, we have deleted original Fig. 4i, Fig. 6e and 6f, and revised the manuscript accordingly.

2. The author added a new allelically marked Treg transfer experiment in Fig. 7j and 7k to show *Senp3*^{-/-} Tregs are less stable in tumor microenvironment. However, they didn't show the percentage of CD45.2⁺ transferred Tregs that lost Foxp3 expression in the tumor. This number is the key data to prove the authors' statement on Treg instability.

Response: We appreciate the reviewer's valuable comments and have included the results showing that the proportion of CD45.2⁺Foxp3⁻ T cells was substantially greater in the tumors of B6.SJL mice injected with SENP3-deficient Treg cells than in those of B6.SJL mice injected with wild-type Treg cells (Fig. 7j), indicating an important role of SENP3 in maintaining the stability of Treg cells. Moreover, NAC treatment increased the percentage of tumor-infiltrating wild-type CD45.2⁺ Foxp3⁻ T cells (Fig. 7j), indicating that treatment with NAC perturbs tumor-infiltrating Treg cell stability.

3. The FACS plots in Fig. 2a do not seem to be correct (it's different from the previous version). For example, 16.2% Foxp3⁺ gate in spleen doesn't seem to match the actual dot plot shown.

Response: We apologize for the mistake in the reproduction of Fig. 2a. We have now revised Fig. 2a.

Response from reviewer 2 regarding concerns of reviewer 1 from previous round of review:

Overall, I feel the revised manuscript largely addressed reviewer 1's critiques for the original version of the manuscript. There are 3 main points from reviewer 1.

1. The authors might be overstating the effect of NAC in suppressing tumor growth. In the revised manuscript, the authors added new evidence showing that NAC's effect on tumor is partly dependent on Treg cells and targets SENP3. They also softened the conclusion significantly compared to the original version, which I think it's acceptable now.

Response: We appreciate the reviewer's support.

2. Some of the RT-qPCR data and FACS cytokine profiles look very similar and seem too good to be true.

The authors repeated a lot of RT-qPCR and FACS experiments shown in Fig. 4 (previously Fig. 5). The current results (Fig. 4f and 4h) look very realistic compared to the results shown in the original version. I did notice that a lot of genes were removed from the RT-qPCR and FACS analysis in Fig. 4 this time. They also completely

removed RT-PCR result in Fig. 8d. There was no mentioning in their response on why those results were removed. Nevertheless, I think the data shown in the revised manuscript is adequate to support their conclusion.

Response: We thank the reviewer for her/his positive evaluation and apologize for the no mentioning in our response on why those results were removed.

For the RT-qPCR data in previous Fig. 5, we repeated the experiment and confirmed the relative expression of selected genes (**Fig. 4f**). We only showed part of selected genes (**Fig. 4f**).

For the FACS data in previous Fig. 5, iTreg cells differentiated *in vitro* is unacceptable to be used for the analysis of Treg cell stability. We followed the reviewer's valuable suggestions to stimulate nTreg cells *in vitro* for the analysis of cytokine expression and deleted FACS data in previous Fig. 5.

For the RT-qPCR data in previous Fig. 8d, expressions of *Il4*, *Il13* and *Il22* in Treg cells treated with NAC or not *in vitro* were shown. We agree with the reviewer that these cytokines are not as important as IFN- γ for anti-tumor immunity *in vivo*. We followed the reviewer's valuable suggestions to include the data of IFN- γ expression in tumor-infiltrating Treg cells (**Fig. 7g,h,k,l**) and deleted RT-qPCR data in previous Fig. 8d.

3. The RNA-seq data analysis is biased.

The authors added fig. 4a and 4b, and explained how they selected the subsets of genes that are differentially expressed in WT vs KO Tregs. They also pointed out that GSEA method was used to narrow down the T effector cell and Treg signature genes shown in Fig. 4c to 4e. I do think their approach is reasonable here.

Response: We appreciate the reviewer's support.

In conclusion, I think the quality of this revision improved quite a bit compared to the original version, which did contain some dubious results. They still have to address the questions I raised last time before I feel comfortable to give my green light.

Response: We thank the reviewer for her/his positive comments on our work and careful evaluation of the manuscript. Following her/his comments, we have included the additional data to address the concerns and carefully revised the manuscripts accordingly. Please see the summary of response to the comments.

1. nTreg cells, not iTreg cells, should be used for the analysis of Treg cell stability in Fig. 4i, 6e,f. We isolated splenic nTreg cells (CD4⁺CD25⁺GFP⁺) from *Rag1*^{-/-} mice reconstituted (for 8 weeks) with *Bach2*^{+/+} *Cd4*-Cre, *Bach2*^{fl/fl} *Cd4*-Cre, *Senp3*^{+/+} *Cd4*-Cre or *Senp3*^{fl/fl} *Cd4*-Cre bone marrow cells transduced with empty vector (EV) or vector expressing WT or 2KR BACH2 for the *in vitro* Treg stability assay. These new data (**Fig. 6e,f**) demonstrated that BACH2 deSUMOylation maintains Treg cell stability. As suggested, we have deleted original Fig. 4i, Fig. 6e and 6f, and revised the manuscript accordingly.

2. The percentage of CD45.2+ transferred Tregs that lost Foxp3 expression in the tumor: We have included the results showing that the proportion of CD45.2⁺Foxp3⁻ T cells was substantially greater in the tumors of B6.SJL mice injected with SENP3-deficient Treg cells than in those of B6.SJL mice injected with wild-type Treg

cells (**Fig. 7j**), indicating an important role of SENP3 in maintaining the stability of Treg cells.

3. The FACS plots in Fig. 2a do not seem to be correct: We apologize for the mistake in the reproduction of Fig. 2a. We have now revised Fig. 2a.

We appreciate the reviewer's positive comments and insightful suggestions that have helped us clarify and strengthen our main findings.

REVIEWERS' COMMENTS:

Reviewer #2 (Remarks to the Author):

In this revision, Dr. Zou and colleagues have addressed the remaining concerns the reviewer raised in the last round. It is this reviewer's opinion that the current manuscript has met the standard for publication in Nature Communication.

Authors' Responses to the Reviewers' Comments:

Reviewer #2 (Remarks to the Author):

In this revision, Dr. Zou and colleagues have addressed the remaining concerns the reviewer raised in the last round. It is this reviewer's opinion that the current manuscript has met the standard for publication in Nature Communication.

Response: We are grateful to this reviewer for his/her enthusiastic support.